# The Effect of Diet and Exercise Interventions on Body Composition in Liver Cirrhosis: A Systematic Review

**DOI:** 10.3390/nu14163365

**Published:** 2022-08-17

**Authors:** Heidi E. Johnston, Tahnie G. Takefala, Jaimon T. Kelly, Shelley E. Keating, Jeff S. Coombes, Graeme A. Macdonald, Ingrid J. Hickman, Hannah L. Mayr

**Affiliations:** 1Department of Nutrition and Dietetics, Princess Alexandra Hospital, Woolloongabba, QLD 4102, Australia; 2Faculty of Medicine, The University of Queensland, Brisbane, QLD 4072, Australia; 3Centre for Online Health, Faculty of Medicine, The University of Queensland, Brisbane, QLD 4072, Australia; 4Centre for Health Services Research, Faculty of Medicine, The University of Queensland, Brisbane, QLD 4072, Australia; 5School of Human Movement and Nutrition Sciences, The University of Queensland, Brisbane, QLD 4072, Australia; 6Department of Gastroenterology and Hepatology, Princess Alexandra Hospital, Woolloongabba, QLD 4102, Australia; 7Centre for Functioning and Health Research, Metro South Health, Brisbane, QLD 4102, Australia; 8Bond University Nutrition and Dietetics Research Group, Faculty of Health Sciences and Medicine, Bond University, Gold Coast, QLD 4226, Australia

**Keywords:** liver cirrhosis, sarcopenia, sarcopenic obesity, nutrition, exercise, body composition

## Abstract

Alterations in body composition, in particular sarcopenia and sarcopenic obesity, are complications of liver cirrhosis associated with adverse outcomes. This systematic review aimed to evaluate the effect of diet and/or exercise interventions on body composition (muscle or fat) in adults with cirrhosis. Five databases were searched from inception to November 2021. Controlled trials of diet and/or exercise reporting at least one body composition measure were included. Single-arm interventions were included if guideline-recommended measures were used (computed tomography/magnetic resonance imaging, dual-energy X-ray absorptiometry, bioelectrical impedance analysis, or ultrasound). A total of 22 controlled trials and 5 single-arm interventions were included. Study quality varied (moderate to high risk of bias), mainly due to lack of blinding. Generally, sample sizes were small (*n* = 6–120). Only one study targeted weight loss in an overweight population. When guideline-recommended measures of body composition were used, the largest improvements occurred with combined diet and exercise interventions. These mostly employed high protein diets with aerobic and or resistance exercises for at least 8 weeks. Benefits were also observed with supplementary branched-chain amino acids. While body composition in cirrhosis may improve with diet and exercise prescription, suitably powered RCTs of combined interventions, targeting overweight/obese populations, and using guideline-recommended body composition measures are needed to clarify if sarcopenia/sarcopenic obesity is modifiable in patients with cirrhosis.

## 1. Introduction

Advanced liver disease is a complex major health problem, impacting more than 1.5 billion individuals worldwide [1]. Cirrhosis is the end stage of chronic liver disease and is characterised by severe hepatic fibrosis with potential impacts on hepatic function. Once patients develop cirrhosis, they are at risk of dying from decompensated liver disease or hepatocellular carcinoma (HCC) [2]. Liver transplantation offers the opportunity to cure both. During the progression to cirrhosis, many aspects of health deteriorate, increasing the risk of malnutrition and loss of muscle mass [3,4], which in turn are associated with adverse outcomes for patients with cirrhosis and those awaiting transplant [5,6].

There are two key issues relating to body composition for people with liver cirrhosis. Firstly, sarcopenia is a condition characterised by a significant depletion of skeletal muscle in combination with low muscle strength and/or physical performance [7]. Sarcopenia is often interrelated with malnutrition [8]. In general, sarcopenia in liver disease literature refers to reduced muscle mass alone, which has a prevalence in cirrhosis between 40–70% [9]. Sarcopenia is associated with increased mortality in patients with cirrhosis, and in those who receive a liver transplant [10]. The second issue is an elevated body mass index in people with cirrhosis. Comorbid sarcopenia with obesity, where low muscle mass may be masked due to excess adiposity, increases the risk of hepatic decompensation and death in patients with cirrhosis [11,12]. Additionally, surgical risk is increased for obese liver transplant recipients [13,14]. The proportion of patients being referred for liver transplant with comorbid obesity is increasing [15]. Interventions to reduce adiposity may ameliorate the severity of their underlying liver disease, but also needs to be considered to improve transplant outcomes. The challenge in achieving weight loss in this patient group is to preserve or increase muscle mass whilst losing fat mass.

The first challenge in addressing low muscle mass and/or high adiposity in patients with cirrhosis is accurately assessing body composition, which can be complicated by fluid retention with ascites and oedema. Triceps skinfold thickness (TSF) and mid-arm muscle circumference (MAMC) appear less affected by fluid overload than other anthropometric measures in this population [16]. While there is evidence that these measures have good intra- and inter-rater reliability for the diagnosis of malnutrition [17], there remain concerns about their reproducibility [18] and their reliability in identifying subtle changes [7]. Recent guidelines have recommended several reference methods for the assessment of body composition in patients with cirrhosis, specifically computerised tomography (CT) and magnetic resonance imaging (MRI) techniques [16,19]. The use of dual-energy Xray absorptiometry (DXA), bioelectrical impedance analysis (BIA), and ultrasound are also supported when CT/MRI are unavailable. These may be more readily available in clinical settings, although the reliability of BIA and some DXA measures may be adversely impacted by fluid retention in decompensated cirrhosis [20,21]. 

It is well known that both diet and exercise have positive effects on health outcomes across multiple chronic health conditions. Low muscle mass and high adiposity are attractive therapeutic targets in advanced liver disease because they may be modifiable through diet and/or exercise interventions. Exercise training is known to reduce the progression, or reverse muscle wasting [22] and has been shown to improve physical function and frailty in cirrhosis [23]. According to current guidelines [16,19], a high protein, high energy diet has been recommended for people with cirrhosis, due to catabolic effects of cirrhosis that can lead to protein degradation and therefore muscle loss. Minimising fasting times, and the inclusion of a late evening carbohydrate rich snack to prevent overnight catabolism have also been recommended [24]. It is still unclear how to accurately estimate energy needs for individuals with cirrhosis who are obese. There have also been several studies exploring the effect of Branched Chain Amino Acids (BCAAs) in this population; however, there has been heterogeneity in BCAA dosage type [25]. While advice about combined diet and exercise in cirrhosis is beginning to appear in more detail in practice guidelines [26], there remains a gap in current knowledge relating to improving body composition in cirrhosis, especially in obese persons. There is currently no comprehensive synthesis of evidence to guide interventions to slow progression or potentially reverse muscle wasting or reduce adiposity for patients with cirrhosis.

Therefore, we aimed to systematically evaluate the evidence on the effect of diet and/or exercise interventions on body composition in adults with cirrhosis, with a particular interest in the impact of these interventions on patients with obesity and liver cirrhosis to determine whether muscle mass can be preserved concurrently with fat loss.

## 2. Materials and Methods

This systematic review followed the Preferred Reporting Items for Systematic Reviews and Meta-analyses (PRISMA) statement [27] (see Appendix A), and the protocol was registered with the international Prospective Register of Systematic Reviews (PROSPERO ID: CRD42020176547). 

### 2.1. Eligibility Criteria

Table 1 summarises the population, intervention, control, outcomes, and study design (PICOS) for the study selection.

### 2.2. Search Strategy

Databases were searched from inception to 15 November 2021 (PubMed, Embase, Web of Science, CINAHL, and CENTRAL). Reference lists of relevant review articles were hand-searched to identify further articles. The strategy utilised a combination of key words and controlled vocabulary combining terms related to liver cirrhosis AND diet/exercise AND intervention/trial (see Appendix A for full search strategy). The final search was de-duplicated using reference management software, Endnote [28]. References were screened in Rayyan [29]. Two reviewers (H.J. and T.T.) independently screened approximately half of the title and abstracts using a screening tool. Twenty studies were piloted with the tool to determine agreement before completing the screening. For potentially eligible articles, full texts were retrieved and independently screened by two of three reviewers (H.J., T.T., or H.M.). Disagreements were resolved by consensus or referral to the third reviewer.

### 2.3. Data Extraction

Extracted data included study authors, publication year, country, population, setting, intervention, control, and body composition outcomes. If data were not available, an attempt to contact authors was made to retrieve information. Data were initially extracted by either of two reviewers (H.J. or T.T.) in a standardised extraction table. Extraction was piloted across three different study designs (RCT, non-randomised controlled trial, and single-arm intervention studies) to ensure consistency. Where present, we extracted body composition change data between study treatment groups. If unavailable, we recorded the within-group change. All extraction was cross-checked by a second reviewer with disagreements discussed to reach consensus.

### 2.4. Quality Assessment

For each included study, risk of bias was assessed independently by two of three reviewers (H.J., T.T., or H.M.) using the Cochrane risk of bias tool (Rob2) [30] for RCTs, and the ROBINS-I tool [31] for non-randomised controlled and single-arm studies. Rob2 evaluates five domains including risk of bias from: randomisation process, deviations from intended interventions, missing outcome data, measurement of the outcome, and selection of the reported result. The ROBINS-I tool evaluates seven domains including risk of bias due to: confounding, participant selection, classification of interventions, deviations from intended interventions, missing data, measurement of outcomes, and selection of the reported result. For the domains considering deviations from intended interventions, where intervention blinding is considered, we allocated ‘some concerns’ rather than ‘serious’ if participants were not blinded. This is due to the nature of diet and/or exercise interventions, where it is often not feasible for intervention allocation blinding. Conflicts were resolved by consensus or a third reviewer. The certainty of the body of evidence based on outcomes using the Grading of Recommendations Assessment Development and Evaluation was not possible due to significant variability in study design, interventions used, outcome measures employed, and statistical methodologies used across the studies.

### 2.5. Data Synthesis 

A meta-analysis was unable to be performed due to variability in study interventions, control groups, tools to assess body composition, and reporting of means and medians across studies. Narrative synthesis was conducted based on type of intervention and body composition measures. Where a study reported on multiple body composition measures the guideline-recommended measures were prioritised in the text results (CT or MRI, followed by DXA, BIA, or ultrasound, then anthropometry).

## 3. Results

### 3.1. Characteristics of Studies

The final search contained 10,099 articles, including three articles from hand searches (Figure 1). A total of 152 full text articles were retrieved and 27 studies included in this review. Thirty-two studies were excluded for not reporting on body composition measures. The characteristics and outcomes of the included studies are summarised in Table 2. Of the 27 studies, 19 were RCTs, 3 were non-randomised controlled trials, and 5 were single-arm intervention studies. Most studies were relatively small, with participant numbers ranging from 6 to 120, totalling 1263 participants. Intervention duration ranged between 4 and 56 weeks and populations included patients with both compensated and decompensated cirrhosis. Only one study specifically targeted an overweight population [32], however the primary outcomes of interest were weight loss and portal hypertension changes. Thirteen studies in total reported populations with a mean BMI either overweight [33,34,35,36,37,38,39,40] or obese [32,41,42,43,44]. Others did not report on BMI [45,46,47,48,49,50,51]. None of the studies specifically targeted sarcopenic obesity in cirrhosis.

For most of the included studies, the change in muscle or fat mass was a secondary outcome, and factors such as muscle strength [29], aerobic/exercise capacity [41,46], survival [39,47], quality of life [48], portal hypertension [28], hepatic venous pressure [30], or liver function [49] were primary outcomes. Fourteen studies were combined diet and exercise interventions [32,33,35,36,37,38,39,40,42,43,44,47,52,53], all in outpatient settings. Their exercise components varied, with most delivering supervised sessions in a clinic setting [32,33,35,36,40,42,43], although one study was supervised by a clinician at the patient’s home [47] and others were self-directed at home [37,38,39,44,52,53]. Most exercise was moderate to high intensity for 30–60 min sessions on 1 to 3 days a week and utilised either aerobic or resistance training, or a combination of these. Otherwise, some self-directed sessions focused on increasing step counts. The dietary component of five of these combined interventions used a high protein and energy diet [35,36,40,42,44]. Four of those studies provided the same dietary intervention to the control group, with the only difference between treatment groups being exercise in the intervention arms [35,36,42,44], while only one study provided “usual care” to the control participants [40]. Another combined study followed this style, however providing ‘standard dietary advice’ to both intervention and control arms, while the intervention arm also received supervised exercise training [43]. Three other combined studies delivered exercise and diet interventions to both groups, with the difference being a specific dietary product, either non-alcoholic beer [37], branched-chain amino acids (BCAAs) [33,39], or beta-hydroxy-beta-methylbutyrate (HMB, a metabolite from leucine) [38]. Of the three diet and exercise single-arm interventions, two provided BCAAs, with self-directed exercise [52,53] while the third study in overweight cirrhotic patients focused on a hypocaloric, moderate protein diet with supervised exercise [32].

Twelve diet-only studies [34,45,48,49,50,51,54,55,56,57,58,59] were included: ten in an outpatient setting [34,48,49,50,54,55,56,57,58,59], one in inpatients [45], and the twelfth commenced in inpatients with outpatient follow up [51]. Most (*n* = 9) interventions prescribed a high protein and energy diet plus oral nutritional supplements either with [48,49,50,56,57,59] or without BCAAs [54,55,58]. Out of the three remaining studies, one prescribed a high energy diet without supplementation [34], one study utilised 3–4 weeks of enteral nutrition follow by oral supplementation [45], and one study utilised short-term parenteral nutrition in combination with a high protein and energy diet [51]. 

One exercise-only intervention met the eligibility criteria. This RCT involved supervised aerobic and resistance exercise sessions of moderate intensity for 60 min three times weekly, versus a relaxation program for the control group of the same frequency and duration [41].

Across all studies, ten different methodologies were used to measure body composition (see Table 2). Most combined diet/exercise interventions used guideline-recommended measures: CT plus DXA and BIA [38], CT plus BIA [46], MRI plus BIA [29], ultrasound [37,41], or BIA alone [28,30,50]. Two diet-only studies used CT [54] or BIA [53], while the exercise-only intervention utilised DXA [51]. Anthropometric measures on their own were used predominantly in diet only studies, with the most frequent variables measured being MAMC in 12 (44%) and TSF in 11 (41%) studies. Some other anthropometric measures including calf and thigh circumference were utilised alongside guideline-recommended measures.

### 3.2. Quality Assessment

Plots summarising the risk of bias are presented in Figure 2 and Figure 3. For the 19 RCTs, high risk of bias was most prevalent in domain 4 (bias in the measurement of the outcome), where assessors were often not blinded to the intervention. Almost all studies were low risk for domain 1 (randomisation and concealment processes). For domain 2, evaluating if participants and/or interventionists were blinded to the intervention allocation, the majority were allocated ‘some concerns’. For the eight non-randomised studies, high risk of bias was most common in domain 3 (classification of interventions), because five of these studies were uncontrolled with no group allocations.

### 3.3. Outcomes for Combined Diet and Exercise Intervention Studies

From the nine combined diet and exercise RCTs, four showed significant improvements in lean mass measured by CT [39], MRI [40], BIA [36], and quadricep ultrasound [35] compared to controls. One study also observed significant reductions in fat mass [35]. Three of these four studies had similar interventions of supervised, moderate intensity exercise (aerobic and/or resistance) on 3 days/week over 8–14 weeks plus targeted protein intakes above 1.2 g/kg/day through either provision of oral nutrition supplements in addition to diet, or dietetic counselling [35,37,40]. The intervention of the fourth diet and exercise RCT [39] that demonstrated an increase in skeletal muscle mass relied on frequent meals plus BCAA supplementation, with the exercise component being an increase in the number of daily steps. The participants in the control arm of this study were exposed to the same diet and exercise intervention, but received a placebo instead of BCAAs, implicating these in the improvement in muscle mass.

A combined diet/exercise RCT [44] which used counselling for self-directed exercise and a high protein diet with BCAA supplementation demonstrated a significant improvement compared to a diet-only control group in psoas muscle index via CT, but not in any other measures of muscle/lean mass (CT, MRI, or DXA). While the intervention group significantly increased daily number of steps compared to the control group, this small study population (*n* = 17) may have limited the power to detect change in some measures. This cohort of transplant candidates also had more advanced liver disease compared to the other combined interventions.

Four of the remaining diet/exercise RCTs reported significant increases in muscle mass, however this was only reported within study groups. These four studies used either supplements in combination with a diet and exercise intervention, (including HMB [38], non-alcoholic beer [37], or the amino acid leucine—a BCAA [33]); or provided dietetic counselling adjusted for BMI categories [47]. While two studies indicated good adherence to the diet and exercise interventions [37,38], the other two did not report adherence [33,47]. 

Both of the non-randomised combined diet/exercise intervention studies found no significant changes in lean or fat mass measured via BIA [42] or MAMC [43]. Both these studies had study population numbers of <40. One had only nine participants complete the intervention (39% attrition rate), so sample sizes may have been too small to identify significant changes [43]. The three single-arm combined diet/exercise studies assessed outpatients with compensated cirrhosis showed mixed results [32,52,53]. An intervention targeting overweight participants (BMI > 26 kg/m^2^) with a 16-week program of supervised exercised with reduced caloric intake observed a significant reduction in fat mass with no significant change in lean mass [32]. A second single-arm combined intervention targeting increased step activity and BCAA supplementation reported a significant increase in muscle volume via BIA, expressed only as change ratio [52]. The final single-arm study reported no significant changes in skeletal muscle via CT, after 12 months of BCAA supplements and prescribed bench step activity [53]. Compliance was not reported, and this study was limited by a small (*n* = 6) all-female cohort. 

### 3.4. Outcomes for Diet-Only Intervention Studies

Three of nine diet-only RCTs found a significant increases in lean mass assessed by MAMC [34,48,56], while another showed a decline in the control group without change in the intervention cohort [51]. Okabayashi et al. [56] demonstrated an increase in MAMC in an intervention group using a carbohydrate enriched BCAA supplement over 12 months, combined with dietary advice to reduce energy intakes to offset the extra energy supplied with the supplement. The aim was to match dietary energy intakes to the control participants who received no supplementation; however, no dietary compliance data were reported. The second RCT [34] observed an increase in MAMC with a 12-week prescribed high energy (35–40 kcal/kg/day), low sodium diet compared to usual diet. This was a two-period cross over trial where two groups followed a prescribed diet and usual diet. The within-group change data indicated both groups significantly increased MAMC after the prescribed diet, while MAMC either declined or remained stable with usual diet. Compliance to the prescribed diet was reportedly high in both groups. The third study was of hospitalised patients. This study utilised BCAAs versus a maltodextrin supplement in the control group [48]. Short-term enteral nutrition was also provided in both groups if hepatic encephalopathy occurred and continued until oral intake was well established. The BCAA group had a significant within-group increase in MAMC, but not a significant change compared to controls. 

The five remaining RCTs of diet-only interventions mostly assessed lean mass using MAMC and found no significant changes with the intervention [45,50,54,55,57]. An RCT [45] of inpatients receiving enteral nutrition for 4 weeks as a component of their intervention found no significant changes in MAMC or TSF compared to inpatients receiving a usual hospital diet. A four week diet-only RCT [57] provided an isocaloric diet for both intervention and controls (2000 kcal and 80 g of protein/day), with the intervention group receiving BCAA supplementation and a reduced diet to achieve the same energy and protein intake as the control. Only within-group changes were reported and no diet compliance data were presented. MAMC did not significantly change in participants given the BCAA supplement over 12 months compared to usual diet [50]. Average intakes declined marginally in both groups even though BCAA compliance was satisfactory. A study in transplant candidates [55] with a MAMC below the 25th percentile also saw no improvement in this measure following supplementation and dietary counselling until transplantation, versus dietary counselling alone. The RCT by Hirsch et al. [54]. provided supplements over 12 months to patients with decompensated cirrhosis. While mean oral intakes appeared significantly higher in the intervention versus control, there were no significant changes in MAMC or TSF. 

The final diet-only RCT [51] evaluated the effect of three diets in people with decompensated cirrhosis and ascites on lean and fat mass assessed by anthropometry. The first diet (Group A) prescribed 24 h of parenteral nutrition in addition to a high energy and protein diet with monthly dietitian advice. Group B received the same diet without parenteral nutrition while the third (control) group were prescribed a “sodium free” diet with dietitian advice. The control group had a significant decline in TSF and MAC compared to Groups A and B. MAMC was not reported in this study. Unfortunately, the control group had mean dietary protein and energy (0.6 ± 3 g/kg/day and 25 ± 8 kcal/kg/day respectively), considerably below guidelines for decompensated cirrhosis [16]. This is likely the cause for the changes in the control group and highlights the potential negative impact of a restrictive low sodium diet without a protein or energy prescription in patients with decompensated cirrhosis.

Three non-randomised or single arm studies of diet-only interventions yielded mixed results [49,58,59]. One non-randomised study [49] assessed patients who had undergone surgery for HCC and reported a within-group increase in MAMC after 6 months of BCAA supplementation twice daily, using a historical control group who received no supplementation. No MAMC data were reported for the control and MAMC was not compared between groups. One of the two single-arm diet-only interventions [58] saw a significant improvement in lean mass via BIA with a soy-based nutritional supplement over 8 weeks plus usual diet. Dietary intake changes were not reported. Participants had predominantly Child-Pugh A cirrhosis, allowing a reasonably reliable interpretation of BIA. The final single-arm diet-only study [59] reported no significant change in skeletal muscle via CT. The intervention of BCAA supplementation over 48-weeks was said to have 100% adherence to the supplement. Intramuscular adipose tissue was also assessed via CT with no significant change observed.

### 3.5. Outcomes for Exercise-Only Interventions

The one exercise-only study [41] was an RCT involving 12 weeks of supervised moderate intensity aerobic exercise 3 days/week, compared to a “sham intervention” of relaxation exercises. The exercise group, which reported high attendance rates, significantly increased lean mass via DXA. Additionally, there was a significant within-group reduction in fat mass in the intervention group as well as an increase in upper thigh circumference and reduction in mid-arm circumference. There were no significant changes in the “sham” group, but comparisons between the active and sham groups were not reported.

## 4. Discussion

The aim of this systematic review was to assess the impact of diet and/or exercise interventions on body composition in patients with liver cirrhosis. While published reviews exist on nutrition and/or exercise interventions in cirrhosis, there are none, to our knowledge, that reviewed studies which specifically measured body composition across both diet and exercise interventions. Secondly, this review also sought to determine the effect of these interventions in patients with cirrhosis and obesity, given the increasing prevalence of obesity and the risks associated with this [12], versus the potential deleterious impact of calorie restriction on muscle mass in this population.

Unfortunately, the 27 studies identified for this review were too heterogeneous in terms of design and outcome measures to allow meta-analysis. Small study size and failure to report adherence to interventions also impacted data synthesis. Nonetheless, on systematic review, the combined diet and exercise interventions appeared to show the greatest potential to increase muscle mass. To demonstrate an increase in muscle mass with exercise, these interventions needed to be of ≥8 weeks duration and comprise 30–60 min of moderate intensity supervised exercise (aerobic and/or resistance), on at least 3 days per week combined with protein intakes of 1.2–2 g/kg/day. In addition, there appeared to be a benefit to muscle mass from BCAA supplementation [35,36,39,40]. Interestingly, several of the combined RCTs [35,37,38,39,44] provided the control group with either a diet or exercise intervention. Based on these studies it appears that there is a synergistic effect when both diet and exercise interventions are delivered to increase muscle mass. 

Obesity is known to impact patients with cirrhosis as an important contributor to progression of liver disease. In patients undergoing liver transplant, severe obesity (BMI > 35 kg/m^2^) increases the risk of peri-transplant complications and death [61]. The prevalence of obesity is increasing in the whole population and in patients with advanced liver disease, and so the impacts of obesity in patients with advanced liver disease are likely to become increasingly important. A challenge addressing obesity in patients with cirrhosis is that the catabolic metabolism found in advanced liver disease could potentially result in significant muscle loss with calorie restriction. Of the 27 studies included in this systematic review, 19 reported on dry weight BMI, with the mean BMI of patients in 13 of these studies being in the overweight [33,34,35,36,37,38,39,40] or obese [32,41,42,43,44] ranges. Only two studies reported on changes in fat mass [32,39]. In the RCT by Hernandez-Conde and colleagues [39], the mean BMI of patients in intervention and control arm were in the overweight range. Although weight loss was not a specific goal of their study, they showed that a combined intervention of diet, exercise, and BCAA supplementation led to a reduction in fat mass while muscle mass improved. The one study targeting weight loss in overweight and obese patients with cirrhosis was promising in that it demonstrated a fall in body weight with maintenance of lean mass after 16-weeks of a combined intervention of exercise with a reduced energy, moderate protein diet [32].

In relation to the heterogeneity of studies, one issue that impacted the ability to synthesise the findings of this review was that 10 different methods of assessing body composition were used across the 27 studies. Current guidelines recommend CT or MRI as optimal body composition assessment methods, in part because they are less impacted by the fluid overload and ascites that occur in decompensated cirrhosis than some of the other methods [16,19]. While CT/MRI are expensive and not always available, they are often part of standard of care for patients undergoing transplant evaluation to assess hepatic vasculature or for HCC monitoring. Although these routine measures are not performed specifically to assess body composition, they can additionally be used to assess muscle and fat mass; and it is possible for allied health clinicians to perform these analyses [62]. 

When abdominal CT or MRI are not available, guidelines recommend using DXA and BIA to assess body composition, on the provision that fluid retention is not an issue [19]; however, this restricts their utility in the group of cirrhotic patients most at risk of sarcopenia, those with decompensated disease. Muscle mass quantification by DXA has been shown to correlate with CT in cirrhosis [63]. Ultrasound is promising yet requires further exploration in this population [6,16]. While the accuracy of BIA can be affected by hydration [21], the use of Phase Angle from BIA may provide a more reliable assessment of nutritional status in cirrhosis than other BIA modalities [19], with results comparable to CT [64]. Several studies only utilised MAMC and TSF as outcome measures, particularly in the diet only interventions. Anthropometry is routinely used in the clinical assessment of nutritional status. However, the utility in clinical studies is less clear as these measures suffer in regard to reliability [18], and cannot distinguish small changes in body composition [7,26]. This makes them less than ideal for studies conducted over 8–12 weeks like a number of the studies reported here. Additional issues with their use in the studies included in this review were that outcome assessors were frequently not blinded to intervention arm, or for these very operator dependent measures, that several assessors may have been involved in the serial measurements. This increases the impact of interobserver variability on findings. Interestingly, while there is a strong body of evidence indicating the deleterious effects of sarcopenia in cirrhosis, very few of the included studies assessed if the patient’s level of baseline muscle mass was indicative of sarcopenia prior to conducting the intervention. This highlights the need for future studies to evaluate baseline muscle mass and therefore sarcopenia to understand the true effect of diet and/or exercise interventions. 

An additional issue was that most diet and exercise studies in cirrhosis have small study populations with body composition measures generally underpowered and as mentioned, were often included as secondary outcomes of the studies. Some of the challenges to increasing participant numbers in studies in this area are the complexities of conducting lifestyle interventions in a population with advanced liver disease who may be quite unwell. Another factor which can impact drop-out rates in this population is inclusion of participants who are potential transplant candidates. In one of the RCTs included in this review [43], a 6-week exercise intervention was completed in just over half (56%) of potential liver transplant recipients, and this was largely because of study participants receiving a liver transplant rather than not adhering to the program. We faced a similar issue in an 8-week pilot feasibility RCT of exercise in patients on a liver transplant waiting list [65], only 50% of participants completed the study, largely because participants received their liver transplant within the study period. 

This review also highlighted the sparsity of relevant intervention studies which have targeted patients with decompensated liver disease. Patients with decompensated disease are a complex and high-risk population, who are more likely to experience muscle wasting and adverse outcomes [6]. Chen et al. [44], is one of the few studies in this review that included only decompensated cirrhosis patients that used a combined diet and exercise intervention. This study was small; however, they were able to demonstrate that home-based exercise is safe in this population. This is promising, and future studies should focus on these populations to better understand how body composition can be improved pre-transplant to improve morbidity and mortality.

While this systematic literature review focused on changes in body composition measured using methodology validated in liver cirrhosis, there are other diet and or exercise intervention studies that have added value to the management of patients with advanced liver disease. Several exercise RCTs in patients with cirrhosis have measured aspects of physical performance including strength, exercise capacity, and/or physical function and therefore did not meet the inclusion criteria for this review. Measures such as hand grip strength, anaerobic threshold by cardiopulmonary exercise testing and functional performance assessments such as the Short Physical Performance Battery and the Liver Frailty Index have demonstrated associations with patient outcomes [5,66,67,68]. They can be useful as screening tools to identify patients at risk of complications [69] and are recommended as part of the evaluation of nutritional status in people with cirrhosis [16,19,26]. Consideration should be given to including these measures in future studies that address body composition alongside functional status.

The field of diet and exercise interventions in patients with cirrhosis is obviously at an early and evolving stage. An important goal for future studies is to determine the significance of modest improvements in body composition both in terms of clinical outcomes, but also in patient-important outcomes and their quality of life. Given the potential range and combination of diet and exercise interventions, defining minimal clinically important differences for muscle and fat mass and thresholds for adverse outcomes patients should be a goal to facilitate comparisons between interventions.

## 5. Conclusions

In summary, effective interventions to improve body composition in cirrhosis appear more likely to succeed if diet and exercise components are combined. There remains a paucity of studies in patients with cirrhosis and obesity despite the increasing prevalence of obesity in this population. At present, the evidence supporting diet and exercise approaches to improve body composition in cirrhosis is impacted by underpowered, short-term interventions. Future research should be directed at appropriately powered combined diet and exercise RCTs of at least 8 weeks duration. Ideally assessments of changes in muscle mass, particularly in patients with decompensated cirrhosis should rely on guideline-recommended methods in this population, specifically CT or MRI. These studies should ideally be large enough to allow for the potentially high rates of patient drop-out and include formal assessments of patient adherence to interventions to identify strategies that do and do not work in this cohort. An important goal for future studies should be to determine what are clinically meaningful changes in body composition in patients with cirrhosis as this will facilitate comparison between intervention strategies. These approaches will help clarify if sarcopenia and sarcopenic obesity are modifiable risk factors in cirrhosis.

## Figures and Tables

**Figure 1 nutrients-14-03365-f001:**
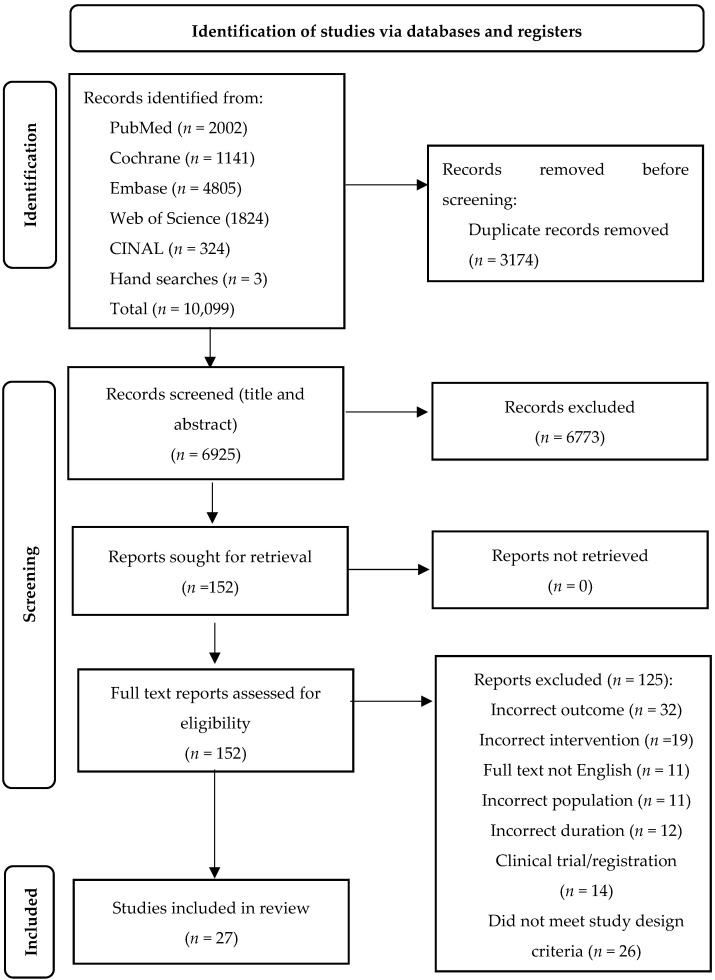
Preferred Reporting Items for Systematic Reviews and Meta-Analyses (PRISMA) flow diagram of the study selection process.

**Figure 2 nutrients-14-03365-f002:**
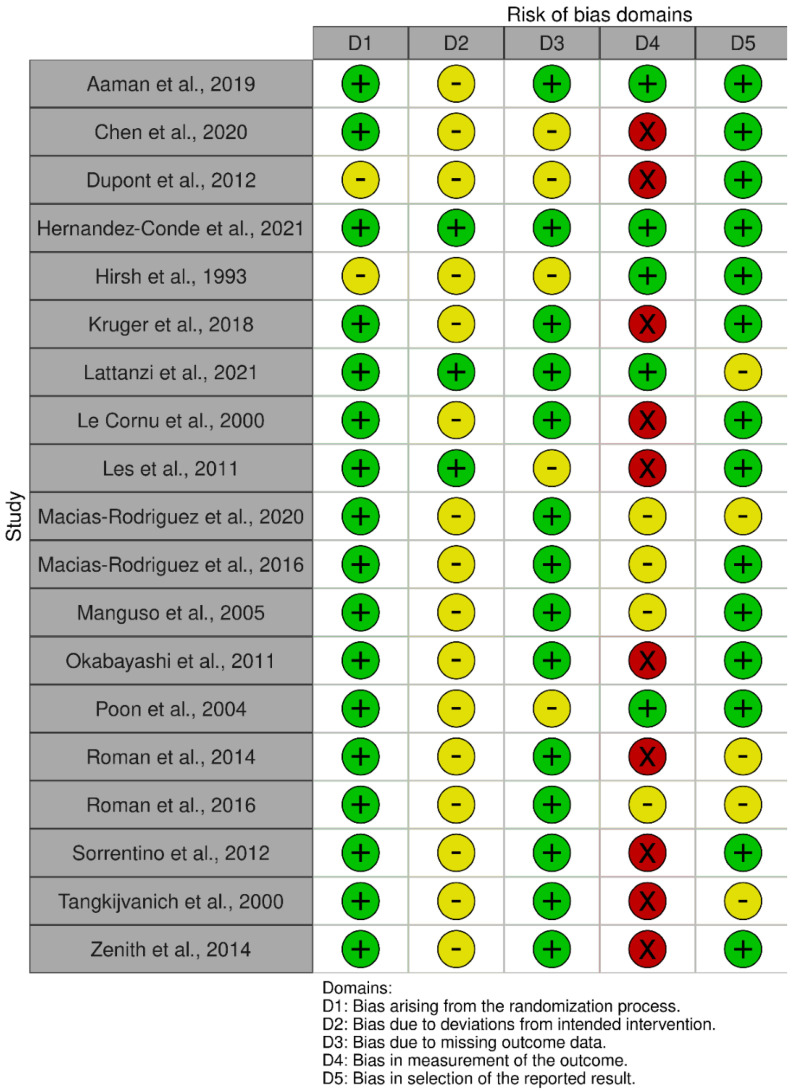
Risk of bias summaries for RCTs using Cochrane Risk of Bias 2 Tool [33,34,35,36,37,38,39,40,41,44,45,47,48,50,51,54,55,56,57].

**Figure 3 nutrients-14-03365-f003:**
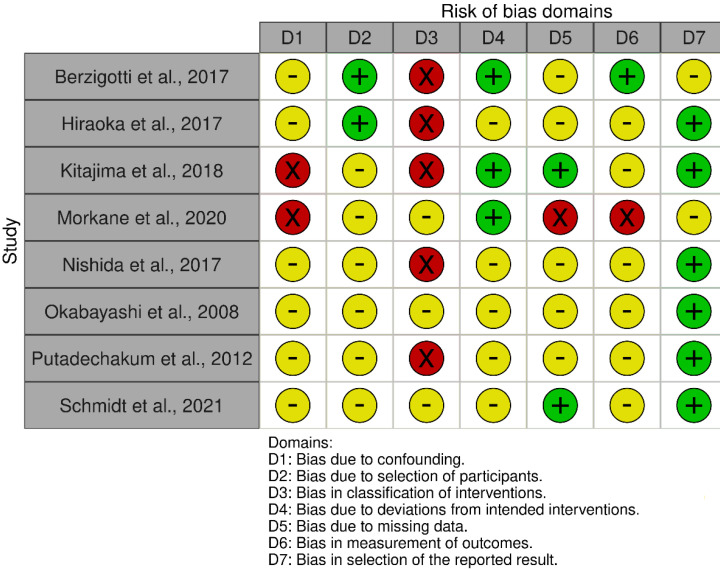
Risk of bias summaries for non-RCTs using Cochrane ROBINS-I (risk of bias tool to assess non-randomised studies of interventions) [32,42,43,52,53,56,58,59].

**Table 1 nutrients-14-03365-t001:** PICOS for study selection and eligibility criteria.

Criteria	Inclusion and Exclusion Details
Population	- Liver cirrhosis, including potential transplant candidates.
Intervention	- Diet or exercise intervention (alone or combination), of at least four weeks duration. - Studies excluded if the intervention was a single nutrient (e.g., vitamin D, omega-3 fatty acid), or nutrition was exclusively administered intravenously without oral nutrition support.
Control	- No specified control. - Studies without a control group were included if they reported specific body composition measures (see below).
Outcomes	- At least one body composition measure, via imaging (CT, MRI, or DXA), BIA, ultrasound, or anthropometry (TSF, MAMC, MAC, thigh, or calf circumference). - Single-arm interventions were included if they had one of the guideline-recommended measures (CT, MRI, DXA [19]; or BIA if in compensated cirrhosis). - Waist circumference was not included due to the confounding effect of any ascites.
Study Design	- RCTs, non-randomised controlled trials and single-arm interventions were eligible. - Articles excluded: case report, letter to the editor, abstract only, or non-English.

CT: computerised tomography, MRI: magnetic resonance imaging, DXA: dual-energy Xray absorptiometry, BIA: bioelectrical impedance analysis, TSF: triceps skinfold thickness, MAMC: mid-arm muscle circumference, MAC: mid-arm circumference, RCT: randomised controlled trials.

**Table 2 nutrients-14-03365-t002:** Study characteristics and outcomes for diet and/or exercise interventions in cirrhosis.

Study Citation, Country	Study Design	Population	Exercise Intervention	Dietary Intervention	Control Group	Body Composition Outcomes *↑ = Significantly Increased or Higher* *↓ = Significantly Decreased or Lower* *↔ = No Significant Difference (Pre/Post or* vs. *Control)*
** *Combined intervention studies (n = 9 RCTs, n = 2 non-randomised studies, n = 3 single arm intervention trials)* **						
Aaman et al. [40] 2019 Denmark	RCT	Intervention *n* = 20 Age 61.7 ± 7.8 years 80% male BMI 26 ± 3.0 kg/m^2^ Child Pugh Class: A 50%, B 50% MELD 10.8 ± 2.7 Control *n* = 19 Age 63 ± 7 years 74% male BMI 25 ± 4.2 kg/m^2^ Child Pugh Class: A 53%, B 47% MELD 10.7 ± 2.8 Outpatients	Supervised resistance training 3 days/week for 60 min at a moderate level. 5 min warm up, then 7 whole body exercises, (3 sets for legs, 2 sets for arms/chest, 1 set lower back, 1 for abdominals), starting at 15–12 repetitions at the start down to 8 by week 12 Duration:12 weeks	Oral nutrition supplements (125 mL, 14.4 g protein and 2.9 g BCAA/100 g) provided if protein intake < 1.2 g/kg/day at baseline	No change to current exercise or diet	*Intervention versus control:* ↑ Cross sectional area of quadriceps via MRI ↑ Body cell mass via BIA ↔ Dry lean mass via BIA ↔ Lean mass via BIA ↔ Calf circumference ↔ MAC ↔ Thigh circumference ↔ Mid arm muscle area ↔ TSF
Chen et al. [44] 2020 USA	Pilot RCT	Intervention *n* = 9 Age 55 ± 7 years 56% male BMI 30 ± 6 kg/m^2^ Child Pugh Class: B 78%, C 22% MELD-Na 16 ± 4 Control *n* = 8 Age 54 ± 11 years 75% male BMI 31 ± 8 kg/m^2^ Child Pugh Class: B 50%, C 50% MELD-Na 19 ±3 Portal hypertension and MELD ≥ 10 Outpatients	Education on exercise, and behavioural counselling bi-weekly for first 8 weeks. Self-directed exercise increasing 500 steps/day weekly to biweekly. Daily to weekly motivational phone calls. Duration: 12 weeks	Standardised diet provided 1.2–1.5 g/kg/day of protein + late evening snack + oral nutrition supplement (6 g essential amino acids) twice a day	Standardised diet (same as intervention group) only	*Intervention versus control:* ↑ Psoas muscle index via CT ↔ Total skeletal muscle index via CT ↔ Intramuscular adipose tissue via CT ↔ Total abdominal adipose tissue via CT ↔ Total thigh muscle volume via CT ↔ Thigh muscle index via CT ↔ Cross sectional area, 50% of femur length via CT ↔ Thigh adipose tissue volume via CT ↔ Fat free mass via DXA ↔ Fat mass via DXA ↔ Lean muscle index via DXA ↔ Lower extremities lean muscle index via DXA ↔ Fat free mass via BIA ↔ Fat mass via BIA ↔ Skeletal muscle mass via BIA ↔ Skeletal muscle index via BIA ↔ Phase angle via BIA
Hernandez-Conde et al. [39] 2021	Pilot, double-blind RCT	Intervention *n* = 15 Age 69 ± 9.7 years 86.7% male BMI 29 ± 4.6 kg/m^2^ MELD 10.7 ± 4.4 Child Pugh Class: A 78.6%, B 21.4% Control *n* = 17 Age 61 ± 9.4 years 88.2% male BMI 26 ± 4.7 kg/m^2^ MELD 11 ± 3.4 Child Pugh Class: A 59%, B 29%, C 12% Compensated outpatients	Personalised exercise instructions with use of accelerometers in wristbands or smartphones to include 5000–10,000 steps/day with gradual increments of 2000–2500 steps/day + moderate intensity exercise in 30-min sessions (goal at least 150 min/week) + verbal reinforcement at reviews. Duration: 12 weeks	Personalised diet recommendations + instructed to eat 7 meals/day including late evening snack plus BCAA supplement 100 g dissolved in 500 mL water throughout the day (15 g protein, 8.5 g fat, 68 g of carbohydrates, 2.61 g of leucine, 1.01 g of isoleucine, and 1.62 g of valine) + verbal reinforcement at reviews	Same exercise and diet recommendations as intervention group except took placebo supplement 100 g dissolved in 500 mL water throughout day (maltodextrin 99.63%) instead of BCAA	*Intervention versus control:* ↑ Skeletal muscle index via CT ↓ % total body fat via BIA ↔ Phase angle via BIA
Kruger et al. [47] 2018 Canada	RCT	Intervention *n* = 20 Age 53 ± 8 years 50% male MELD 9.05 Child Pugh Class: A 70%, B 30% Control *n* = 18 Age 56.4 ± 8.5 years 65% male MELD 9.7 Child Pugh Class: A 70%, B 30% BMI not reported Outpatients	Supervised at home, moderate to high intensity aerobic exercise (60–80% of heart rate reserve) on cycle ergometer 3 days/week (30 min sessions gradually increased to 60 min). Visited bi-weekly for session observation. Duration: 8 weeks	Dietary counselling on optimal protein (1.2–1.5 g/kg/day, ideal body weight for BMI > 30) and energy intake (35–40 kcal/kg for BMI 20–30, 25–35 kcal/kg for BMI 30–40, and 20–25 kcal/kg for BMI > 40. Advised on exercise days to consume an extra 250–300 kcal.	Usual care	*Intervention versus control:* ↔ Thigh muscle mass via ultrasound ↔ Thigh circumference
Lattanzi et al. [38] 2021	Pilot single blind RCT	Intervention *n* = 14 Age: 59.2 ± 8.4 years 64% male BMI 29.8 ± 4.3 kg/m^2^ Child Pugh Class: A 86%, B 14% MELD 9 ± 2.7 Control *n* = 10 Age: 56 ± 4.6 years 60% male BMI 29.6 ± 6.8 kg/m^2^ Child Pugh Class: A 90%, B 10% MELD 9.8 ± 3.2 Outpatients with portal hypertension	Motivational interviewing with information on physical activity at baseline	Motivational interview at baseline with information and counselling on diet in line with EASL clinical guidelines (2019) + HMB supplement (3 g/day)	Same exercise and diet as intervention group + placebo supplement (Sorbitol 3 g/day)	*Within group changes:* ↑ Thigh muscle thickness via ultrasound ↔Fat free mass via BIA ↔Phase Angle via BIA
Macias-Rodriguez et al. [37] 2020	RCT	Intervention *n* = 22 Age 53.5 ± 7.6 years 47% male BMI 29.8 ± 4.8 kg/m^2^ Child Pugh Class: A 82%, B 18% MELD 8.5 (7–10) Control *n* = 21 Age 53.7 ± 8.2 years 43% male BMI 29.2 ± 3.7 kg/m^2^ Child Pugh Class: A 95%, B 5% MELD 8 (7.5–9.5) Compensated cirrhosis, outpatients	Given wrist-worn accelerometer as activity tracker. Aim to gradually increase physical activity to reach >2500 steps/day above baseline. Total 5000 steps/day. Light to moderate intensity. Duration: 10 weeks	Harris–Benedict equation was utilised to calculate energy requirements + 10% extra for thermic effect of food and 20% extra for exercise. Diet 60% carbohydrates, 1.3–1.5 g protein/kg/day + remainder from fats + 1.5–2 g sodium restriction/day restriction + non-alcoholic beer at lunch (330 mL/day)	The same diet and exercise prescribed as intervention group without non-alcoholic beer (given a 330 mL bottle of water instead)	*Within group changes:* ↔ Phase Angle via BIA ↑ Thigh circumference ↔MAMC ↔TSF
Macias-Rodriguez et al. [36] 2016 Mexico	Pilot open RCT	Intervention *n* = 13 Age 53 (48–55) years 69% male BMI 27.5 (22.4–28.9) kg/m^2^ Child Pugh score 6 (5–7) MELD 9 (8–12) Control *n* = 12 Age 51 (38–57) years 83% male BMI 27.4 (25–30) kg/m^2^ Child Pugh score 6 (5–7) MELD: 12 (7–14) Compensated outpatients	Supervised exercise 3 days/week of 60–70% max heart rate, for 40 min of aerobic training using cycle ergometer + kinesiotherapy/rhythmic activities) Duration: 14 weeks	Instructed to consume 30% extra calories (65% carbohydrates, 1.2 g/kg/day protein) + no added salt diet of 1.5–2 g/day	Same recommendations as intervention; consume 10% extra calories (65% carbohydrates, 1.2 g/kg/day protein) + no added salt diet of 1.5–2 g/day. Continue regular activities, no new exercise	*Intervention versus control:* ↑ Phase angle via BIA
Roman et al. [33] 2014 Spain	Pilot RCT	Intervention *n* = 8 Age 65.5 (46–72) years 62% male BMI 26.7 (18.3–34.7) kg/m^2^ Child Pugh Class: A 87%, B 13% MELD 9.5 (7–12) Control *n* = 9 Age 61 (43–75) years 78% male BMI 27.6 (19.5–35.3) kg/m^2^ Child Pugh Class: A 78%, B 22%, MELD 9 (7–13) Outpatients with a previous episode of decompensation	Supervised exercise 3 days/week, moderate intensity (60–70% max heart rate) for 60 min. Cycle ergometry and treadmill walking Duration: 12 weeks	10 g oral leucine supplementation daily	10 g oral leucine supplementation daily, no exercise recommendations	*Within group changes:* ↑ Lower thigh circumference (intervention compared to baseline, ↔ control) ↔ Mid or upper thigh circumference (intervention or control) ↔ MAMC (intervention or control) ↔ Mid-arm circumference (intervention or control) ↔ TSF (intervention or control)
Zenith et al. [35] 2014 Canada	RCT	Intervention *n* = 9 Age 56 ± 8 years 78% male BMI 27.7 ± 3.8 kg/m^2^ Child Pugh score: 6.2 ± 1.4 MELD 9.7 ± 2.4 Control *n* = 10 Age 59 ± 6 years 80% male BMI 28.9 ± 4.1 kg/m^2^ Child Pugh score: 6.3 ± 1.4 MELD 10.2 ± 1.9 Outpatients, Child Pugh A or B	Supervised exercise 3 days/week, 60–80% of peak VO_2_, 30 min session, increased by 2.5 min per session each week, 5 min warm up and cool down using cycle ergometer Duration: 8 weeks	Baseline dietetic counselling to reach 1.2–1.5 g/kg of protein (for BMI > 30 adjustments made based on ideal body weight), calories BMI specific (between 14 up to 30 kcal/kg) and instructed to consume an extra 250–300 calories on exercise days	Baseline counselling by dietitian (same as intervention) but no formal exercise regimen	*Intervention versus control:* ↑ Quadricep muscle thickness via ultrasound ↑ Thigh circumference
Morkane et al. [43] 2020 United Kingdom	Non-randomised controlled trial	Intervention *n* = 16 Age 55.6 ± 7.8 years 87.5% male MELD 13.7 ± 4.6 BMI 30.9 ± 5.6 kg/m^2^ Control *n* = 17 Age 55.6 ± 7.8 years 82.7% Male MELD 13.2 ± 3.7 BMI: 27 ± 4.6 kg/m^2^ Outpatients, transplant candidates	Supervised 40 min interval training on cycle ergometer (4–6 × 3 min intervals at 80% of AT (moderate intensity) and 4–6 × 2 min intervals at 50% of difference between VO_2_ at peak and VO_2_ at AT (‘severe’ intensity) with 5 min warm up and cool down) Duration: 6 weeks	Standardised nutrition assessment and advice by transplant dietitian at baseline and 6 weeks	Standard care, no initiation of exercise. Standardised nutrition assessment and advice by transplant dietitian at baseline and 6 weeks	*Within group changes:* ↔ Mid-arm circumference (intervention or control) ↔ MAMC (intervention or control)
Schmidt et al. [42] 2021	Non-randomised controlled trial	Intervention *n* = 11 Age 56.6 ± 9.9 years 63.6% male BMI 30.3 ± 5.4 kg/m^2^ Child Pugh Class: A 91%, B 9% Control *n* = 22 Age 58.7 ± 12.9 years 59.1% male BMI 32.4 ± 5.1 kg/m^2^ Child Pugh Class: A 86%, B 14% MELD—not reported Compensated outpatients	Supervised exercise 3 days/week, aerobic, moderate intensity (5 min warm up, 30 min walking/running 60–70% VO_2_ max). Increasing session by 2 min until reaching 50 mins by week 8. Duration: 12 weeks	Diet advice to aim for 25–30 kcal/day and 1.2–1.5 g of protein/kg/day—using estimated dry body weight.	The same diet advice without any exercise intervention	*Intervention versus control:* ↔ Phase Angle via BIA ↔ Lean mass via BIA ↔ Fat mass via BIA ↔ MAMC ↓ MAC
Berzigotti et al. [32] 2017 Spain	Multi-centre single arm intervention pilot study	Total *n* = 50 Age 56 ± 8 years 62% male BMI 33.3 ± 3.2 kg/m^2^ MELD 9 ± 3 Child Pugh Class: A 92%, B 8% Compensated outpatients with BMI ≥ 26 kg/m^2^	Supervised exercise 1 day/week for 60 min moderate intensity (10–12 Borg Scale of Perceived Effort) in groups of 1–5 + increase daily step activity Duration: 16 weeks	Reduction of 500–1000 kcal/day. Protein intake maintained at 20–50% of total kcal and within 0.8 g/kg ideal bodyweight/day. Carbohydrates 45–50% and fat <35% of total kcal. 20 g/day alimentary fibre recommended.	No control	↓ Fat mass via BIA ↔ Lean mass via BIA
Hiraoka et al. [52] 2017 Japan	Single arm intervention study	Total *n* = 33 Age 67 (63–71) years 39% men BMI 23.2 (20.8–25.1) kg/m^2^ Child Pugh Class: A 90%, B 10% Compensated outpatients	Walking (an additional 2000 steps/day on top of usual average steps) Duration: 12 weeks	Late evening BCAA supplement provided once daily (13.5 g protein, 210 kcal/day)	No control	↑ Muscle volume via BIA (reported as change ratio)
Nishida et al. [53] 2017 Japan	Single arm intervention study	Total *n* = 6 Age from 51–79 years 100% female BMI 24.3 (19.6–26.1) kg/m^2^ Child Pugh Class: A 100% Compensated outpatients	Instructed to undertake bench step activity at anaerobic threshold level at home. Aim 140 min/week. Duration: 12 months	BCAA supplement (3 sachets/day = 12.45 g of BCAA), no specific nutrition advice except to maintain usual dietary intake	No control	↔ % fat via BIA ↔ Visceral fat area via CT ↔ Intramuscular adipose tissue content via CT
** *Diet-only intervention studies (n = 9 RCTs, n = 1 non-randomised study, n = 2 single arm interventions)* **						
Dupont et al. [45] 2012 France	Multi-centre RCT	Intervention *n* = 44 Age 56.1 ± 9.6 years 68% male Child Pugh score: 11.2 ± 1.3 Control *n* = 55 Age 54.6 ± 9.6 years 64% male Child Pugh score: 10.5 ± 1.5 BMI or MELD—not reported Inpatients with ARLD and jaundice (without alcoholic hepatitis)	NA	Enteral nutrition 3–4 weeks (30–55 kcal/kg/day through nasogastric tube). Subsequent 3 oral nutrition supplements/day for 2 months Duration: 12 weeks with outcomes reported at 12 months	Standard hospital oral diet	*Intervention versus control:* ↔ MAMC ↔ TSF
Hirsh et al. [54] 1983 Chile	RCT	Intervention *n* = 26 Age 49.9 ± 8.6 years 81% male Control *n* = 25 Age 46.1 ± 8.0 years 84% male BMI, Child Pugh, or MELD—not reported Decompensated outpatients	NA	1 L oral nutrition supplement /day (1000 kcal, 34 g protein) + usual diet Duration: 12 months	Placebo tablet daily	*Intervention versus control:* ↔ TSF ↔ Mid-arm circumference
Le Cornu et al. [55] 2000 England	RCT	Intervention *n* = 42 Age 52 (27–67) years 69% male Child Pugh Class: A 7%, B 48%, C 45% Control *n* = 40 Age 50 (24–68) years 78% male Child Pugh Class: A 10%, B 28%, C 62% BMI or MELD not reported Outpatient transplant candidates with MAMC < 25% percentile	NA	Oral nutrition supplement of 500 mL/day (750 kcal, 20 g protein) was given + dietary counselling to adapt/increase their calories and protein based on their medical condition until transplantation Duration: until transplantation. Median wait 77 (1–395) days intervention and 45 (1–424) control	Standard dietary advice to adapt/increase their calories and protein based on their medical condition until transplantation	*Intervention versus control:* ↔ MAMC ↔ Mid-arm circumference ↔ TSF
Les et al. [48] 2011 Spain	Multi-centre RCT	Intervention *n* = 58 Age 64.1 ± 10.4 years 78% male Child Pugh 8.3 ± 2.0 MELD 16.1 ± 4.5 Control *n* = 58 Age 62.5 ± 10.4 years 74% male Child Pugh 8.1 ± 1.7 MELD 16.2 ± 3.9 BMI—not reported Outpatients with previous episode of hepatic encephalopathy	NA	Diet of 35 kcal/kg + 0.7 g/kg of protein/day adjusted to ideal weight + late evening BCAA supplement 2/day (120 kcal). Enteral nutrition if admitted for episode of hepatic encephalopathy and oral intake in hospital was poor. Duration: mean 32 ± 22 weeks intervention and 36 ± 2 weeks control	Same diet but with maltodextrin supplement 2/day instead of BCAA. Enteral nutrition provided if episode of hepatic encephalopathy and oral intake was poor	*Within group changes:* ↑ MAMC (intervention compared to baseline) ↔ MAMC (control compared to baseline)
Manguso et al. [34] 2005 Italy	Random-ised, double period cross-over trial	Group 1: *n* = 45 Age 60 ± 9 years 67% male BMI 28.5 ± 3.2 kg/m^2^ Child Pugh Class: A 33%, B 77% Group 2: *n* = 45 Age 60 ± 7 years 49% male BMI 27.8 ± 2.1 kg/m^2^ Child Pugh Class: A 33%, B 77% Outpatients with HCV cirrhosis	NA	Group 1: Prescribed diet of 30–40 kcal/kg/day based on calculated desirable weight (total calories split into 16% protein, 55% carbohydrates, 28–30% fat) + low sodium 1000 mg/day Followed by usual diet after. Group 2: Usual diet first. Followed by prescribed diet second. Duration: 3 months per diet (6 months total)		*Within group changes:* ↑ MAMC (Group 1 at 3 months post prescribed diet vs baseline) ↑ MAMC (Group 2 at 6/12, post prescribed diet vs baseline and vs 3/12) ↓ MAMC (Group 1 at 6 months post usual diet vs 3 months post prescribed diet) ↔ MAMC (Group 2 at 3 months post usual diet vs baseline) ↔ TSF (Group 1 or Group 2 after both diet interventions at 3 and 6 months)
Okabayashi et al. [56] 2011 Japan	RCT	Intervention *n* = 40 Age 68 ± 7.6 years 28% male BMI 23.6 ± 3.2 kg/m^2^ Child Pugh Class: A 70%, B 30% Control *n* = 36 Age 65.1 ± 11.3 years 31% male BMI 22.7 ± 3.2 kg/m^2^ Child Pugh Class: A 71%, B 29% Outpatients with scheduled HCC surgery	NA	Carbohydrate and BCAA enriched supplement morning and night. (420 kcal, 13 g free amino acids, 13 g of gelatine hydrolysate, 62 g carbohydrates, 7 g lipids) + dietitian education to modify intake to reduce 420 kcal/day to account for the supplement and match caloric intake to controls Duration: supplements for at least 6 months, with a follow up at 12 months	Usual diet. No supplements	*Intervention**versus control:* ↑ MAMC (at 6, 8, 10, 12 months) ↔ TSF no change post-operatively in both groups (data not reported)
Poon et al. [50] 2004 China	RCT	Intervention *n* = 41 Age 59 (24–84) years 95% male Control *n* = 43 Age 59 (27–80) years 90% male. No BMI, Child Pugh or MELD reported. Outpatients with unresectable HCC	NA	BCAA supplement morning and night (420 kcal, 13 g amino acids, 13 g peptides, 62 g carbohydrates, 7 g lipids) + unrestricted diet unless HE—protein was restricted Duration: 1 week prior to surgery, up to 12 months	Usual diet	*Intervention versus control:* ↔ Mid-arm circumference ↔ TSF
Sorrentino et al. [51] 2012 Italy	RCT	Group A: *n* = 40 Age 64 ± 6.3 years 65% male Child Pugh Class: B 28%, C 72% MELD 12.1 ± 0.7 Group B: *n* = 40 Age 66 ± 7.5 years 67% male Child Pugh Class: B 30%, C 70% MELD 11.7 ± 0.7 Group C: *n* = 40 Age: 65 ± 7.6 years 70% male Child Pugh Class: B 25%, C 75% MELD 12.4 ± 0.9 BMI not reported In/outpatients with refractory ascites	NA	Group A: Instructed to consume 1–1.3 g protein/kg/day, 30–35 kcal/kg/day + low sodium diet (80 mEq/day) + BCAA evening snack (210 kcal, 13.5 g protein, 3.5 g fat) + instructed to adjust energy intake to account for BCAA supplement + post LVP parenteral nutrition for 24 hrs post paracentesis during hospital admission + Dietitian advice monthly. Group B: same as group A without parenteral nutrition post paracentesis. Duration: 12 months, follow up at 3, 6, 12 months	Group C: Low sodium diet (80 mEq /day) + Dietitian advice monthly	*Between group changes:* ↓ TSF (Group C versus Group A at 3, 6, and 12 months and Group C versus Group B at 6 months only) ↓ MAC (Group C versus Group A and Group B at 6 and 12 months)
Tangkijvanich et al. [57] 2000 Thailand	RCT	Group 1: *n* = 14 Age: 53 ± 11 years 71% male BMI 23.7 ± 3.4 kg/m^2^ Child Pugh score: 5–7: 64%, score 8–15: 36%. Group 2: *n* = 15 Age: 53 ± 13 years 80% male BMI: 25 ± 4.1 kg/m^2^ Child Pugh score: 5–7: 60%, score 8–15: 40% Outpatients	NA	Group 1: received standard diet (40 g protein/day) + 150 g BCAA supplement/day = total of ~2000 kcal/day. Duration: 4 weeks	Group 2: standard diet (80 g protein/day = total of ~2000 kcal/day)	*Within group changes:* ↔ MAMC (Group 1 or Group 2)
Okabayashi et al. [49] 2008 Japan	Non-randomised study with historical control group	Intervention *n* = 13 Age 66.2 ± 9.1 years 54% male Child Pugh Class: A 77%, B 23% Control *n* = 28 Age 65.6 ± 8.2 yrs 75% male Child Pugh Class: A 82%, B 18% BMI not reported Outpatients for HCC surgery	NA	Carbohydrate and BCAA enriched supplement morning and night. (420 kcal, 13 g free amino acids, 13 g gelatin hydrolysate, 62 g carbohydrates, 7 g lipids) Duration: 2 weeks prior to surgery and at least 6 months post	Usual care—no supplementation	*Within group changes:* ↑ MAMC (baseline to 6 months for intervention, not reported for control)
Kitajima et al. [59] 2018 Japan	Single arm intervention study	Total *n* = 21 Age 71.3 ± 7.9 years 42% male BMI 23.9 ± 4.0 kg/m^2^ Child Pugh Class: A 48%, B 52% MELD—not reported Outpatients with hypoalbuminaemia	NA	BCAA supplement 3/day after meals. Dietitian advised intakes of 25–35 kcal/kg/day and protein 1–1.4 g/kg/day. Adherence monitored monthly. Duration: 48 weeks	No control	↔ Skeletal muscle index via CT ↔ Intramuscular adipose tissue content via CT ↔ Subcutaneous fat area via CT ↔ Visceral fat area via CT
Putadechakum et al. [58] 2012 Thailand	Single arm intervention study	*n* = 22 Age 52.9 ± 12.8 years 55% male BMI 21.4 ± 0.6 kg/m^2^ Child Pugh Class: A 63%, B 23%, C 14% Outpatients with ARLD	NA	20 g protein (soy based) oral nutrition supplement daily (420 kcal, 20 g protein, 65 g CHO, 10.6 g fat) + regular diet. Duration: 8 weeks	No control	↑ Lean mass via BIA ↔ Fat mass via BIA ↔ TSF
** *Exercise only intervention (n = 1 RCT)* **						
Roman et al. [41] 2016 Spain	RCT	Intervention *n* = 14 Age 62 ± 2.4 years 71% male BMI 31.5 ± 1.6 kg/m^2^ Child Pugh score: 5.4 ± 0.2 MELD 8.2 ± 0.4 Control *n* = 9 Age 63.1 ± 2.3 years 85% male BMI 30.3 ± 1.4 kg/m^2^ Child Pugh score: 5.4 ± 0.2 MELD 9.1 ± 0.4 Outpatients with a previous episode of decompensation	Supervised exercise 3 days/week, 60 min of cycle ergometry and treadmill walking + 5–10 min of upper body resistance exercise + 10–15 min balance, coordination, stretching and relaxation. Moderate intensity (60–70%) of max heart rate. Duration: 12 weeks	NA	Sham intervention 1 h 3 days/week of cephalocaudal muscle relaxation, and breathing, visualisation, and concentration exercises	*Within group changes:* ↑ Lean appendicular mass via DXA (intervention compared to baseline, ↔ control) ↑ Lean leg mass via DXA (intervention compared to baseline, ↔ control) ↑ Lean body mass via DXA (intervention compared to baseline, ↔ control) ↓ Fat body mass via DXA (intervention compared to baseline, ↔ control) ↑ Upper thigh circumference (intervention compared to baseline, ↔ control) ↔ Lower thigh circumference (intervention or control) ↓ Mid-arm circumference and mid-arm skinfold thickness (intervention compared to baseline, ↔ control) ↓ Mid-thigh skinfold thickness (intervention compared to baseline, ↔ control) ↔ MAMC (intervention or control)

Outcome data presented for controlled trials are the between group differences (where reported) and the within group differences if the significance of between group data were not reported. Data presented as mean SD or median (range/inter-quartile range). RCT: randomised controlled trial, AT: anaerobic threshold, MELD: model for end-stage liver disease, BMI: body mass index, ARLD: alcohol related liver disease, BCAA: branched-chain amino acid, CT: computed tomography, DXA: dual-energy X-ray absorptiometry, BIA: bio-electrical impedance analysis, MAMC: mid-arm muscle circumference, TSF: triceps skinfold thickness, MAC: mid arm circumference, HE: hepatic encephalopathy, LVP: large volume paracentesis, EASL; European Association of the Study of the Liver, NA: not applicable, VO_2_ max: maximum amount of oxygen your body is able to use during exercise. Child Pugh score [60].

## Data Availability

The data that support the findings of this study are available from the corresponding author upon reasonable request.

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
