# Peer review of "The Effect of Diet and Exercise Interventions on Body Composition in Liver Cirrhosis: A Systematic Review"

_nutrients, 2022, doi:10.3390/nu14163365_

Round 1

Reviewer 1 Report

This is a very well-written paper.  The methodology is well-designed and presented.  The discussion of results is thoughtful and appropriate. 

Recommended content revisions:

-A large part of the Introduction and some of the Discussion focus on sarcopenia.  However, in the included studies it is not clear if the patients were sarcopenic. Suggest to revise Introduction and appropriate parts of Discussion to focus on the effect of diet and exercise on body composition of patients with cirrhosis.

-In Introduction, focus more on current knowledge on the role of diet and exercise in cirrhosis.

Recommended editorial changes:

-Page 2, second paragraph - change "remains concerns" to "remain concerns"

-Page 13, first paragraph - remove period at the beginning of the paragraph.  Before paragraph, add subtitle for combined diet and exercise RCTs to be consistent with following sections.

-Page 13 and on - fix margins

-Page 15, last paragraph - change to "heterogeneous"

-Page 16, first paragraph - sentence starting with "Several RCTs..." seems out of place

-Page 17, first paragraph - start new paragraph at "An additional issue..."

Author Response

Please see the attachment for the response to reviewer 1. 

Reviewer 2 Report

To the authors:

This is a much-needed study summarizing all the existing information on exercise in cirrhosis and its influence on body composition, the methodology is appropriate, and all the available studies were included. I believe this manuscript will be of great use for its readers.

My only suggestion for the edition of the manuscript, if this is feasible for the journal, is to pull together supplementary figures S1 and S2 to make it into a single figure, and to add it to the main document, not as supplementary material, given the importance of the data.

Author Response

Please see the attachment for reviewer two responses.
